# VITA-Audio: Fast Interleaved Cross-Modal Token Generation for Efficient Large Speech-Language Model

**Zuwei Long**[1,†], **Yunhang Shen**[1,†,♠], **Chaoyou Fu**[2,⋆], **Heting Gao**[1], **Lijiang Li**[2]

**Peixian Chen**[1], **Mengdan Zhang**[1], **Hang Shao**[1], **Jian Li**[1], **Jinlong Peng**[1]

**Haoyu Cao**[1], **Ke Li**[1], **Rongrong Ji**[3], **Xing Sun**[1,⋆]

[1]Tencent Youtu Lab, [2]Nanjing University, [3]Xiamen University
[†] Equal Contribution   [♠] Project Leader   [⋆] Corresponding Author

https://github.com/VITA-MLLM/VITA-Audio

## Abstract

With the growing requirement for natural human-computer interaction, speech-based systems receive increasing attention as speech is one of the most common forms of daily communication. However, the existing speech models still experience high latency when generating the first audio token during streaming, which poses a significant bottleneck for deployment. To address this issue, we propose VITA-Audio, an end-to-end large speech model with fast audio-text token generation. Specifically, we introduce a lightweight Multiple Cross-modal Token Prediction (MCTP) module that efficiently generates multiple audio tokens within a single model forward pass, which not only accelerates the inference but also significantly reduces the latency for generating the first audio in streaming scenarios. In addition, a four-stage progressive training strategy is explored to achieve model acceleration with minimal loss of speech quality. To our knowledge, VITA-Audio is the first multi-modal large language model capable of generating audio output during the first forward pass, enabling real-time conversational capabilities with minimal latency. VITA-Audio is **fully reproducible** and is trained on open-source data **only**. Experimental results demonstrate that our model achieves an inference speedup of $3 \sim 5\times$ at 7B parameter scale, but also significantly outperforms open-source models of similar model size on multiple benchmarks for automatic speech recognition (ASR), text-to-speech (TTS), and spoken question answering (SQA).

## 1 Introduction

Real-time speech systems have become a crucial research focus for enabling natural dialogue. Traditional speech systems predominantly adopt a modular design that decomposes real-time speech processing into three discrete components: automatic speech recognition (ASR), large language models (LLMs), and text-to-speech (TTS) [54, 32, 69]. However, this cascaded approach suffers from cumulative latency, loss of paralinguistic information (*e.g.*, emotional prosody, rhythm) during modality conversion, and error accumulation between modules, substantially lowering the practical utility of cascaded architectures in real-time interactive scenarios. To address the limitations of traditional methods, many recent studies have adopted an end-to-end approach to handle inputs and outputs of the model [58, 10, 23]. These methods directly input speech into LLMs through an audio encoder and then synthesize speech response with discrete tokens [66] or LLM hidden states [56].

39th Conference on Neural Information Processing Systems (NeurIPS 2025).

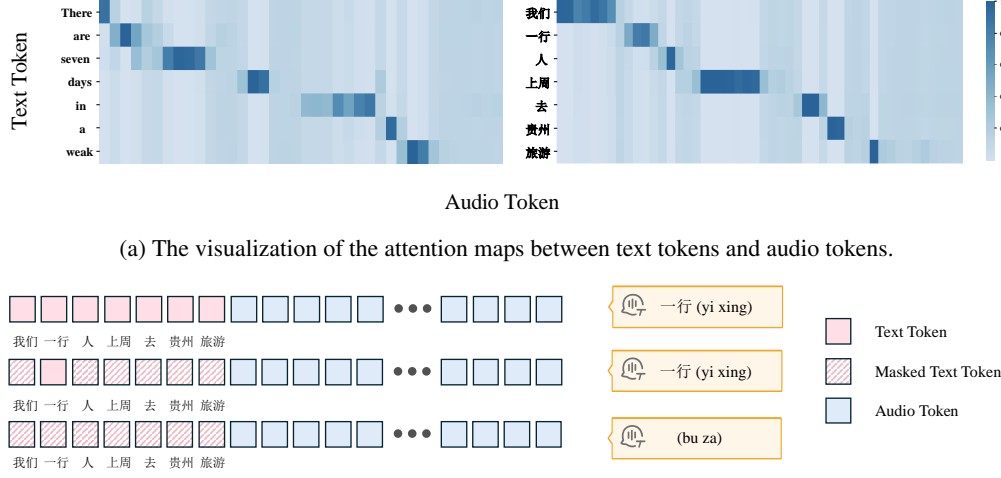

(a) The visualization of the attention maps between text tokens and audio tokens.

(b) The transcription results of the generated audio into text under different attention masks.

Figure 1: (a) The audio sequence generated by the speech model exhibits a strong correlation with the corresponding text tokens. (b) With irrelevant text tokens being masked out, the model is still able to generate the correct audio, and the pronunciation remains contextually appropriate. However, if all text tokens are masked, the model outputs random audio. This suggests that the hidden states from the LLM include sufficient contextual information for generating the corresponding audio tokens. Consequently, the mapping from text hidden states to audio tokens is accomplished using relatively simple modules, without the need for the extensive semantic modeling typically required by LLMs.

While existing end-to-end speech models generate output in a streaming fashion to reduce the response latency, their first token delay is still high. Specifically, current speech models cannot directly deliver the first streaming audio chunk upon completing the first LLM forward pass, *i.e.*, decoding the first text token. In the applications requiring high real-time performance, this delay poses a significant bottleneck to the deployment of LLMs for speech processing. This prompts a pertinent question:

*How can we achieve more real-time audio generation within end-to-end speech models?*

To explore this issue, we visualized the hidden states of the final decoder layer of the speech model. As shown in Fig. 1a, the audio tokens generated by the speech model show increased attention to the text tokens they correspond to. As the generation of audio tokens progresses, the text tokens attended by the new audio token advance accordingly. This finding is also reported in many literature on attention-based speech systems [7, 34]

In Fig. 1b, we show a Chinese sentence with a homograph "行" as an example. The pronunciation of this character can be /xing/ or /hang/ depending on its context. In Chinese, 'yihang' emphasizes spatial arrangement (e.g. a row/line of egrets), while 'yixing' focuses on the concept of a group or unit (e.g. a group/party of people). Our speech model correctly decides the character's pronunciation to be the former, given the hidden states of historical inputs. We then modify the model inference process by masking out all text hidden states, except the one corresponding to the token "一行"(/yixing/) before generating its corresponding audio tokens. This modification prevents subsequently generated audio tokens from directly attending to other text tokens, although they can still attend to previously generated audio tokens. We find that the subsequently generated audio tokens accurately produce the sounds as "yixing", which remains contextually appropriate. The same observation also holds for other non-homograph tokens. We therefore argue that **the hidden states from the LLM include sufficient contextual information for generating its corresponding audio tokens, and attending to additional texts is unnecessary**. Finally, we experiment with masking out all text tokens. This time, the generated audio fails to align with its text and sounds like random non-speech babbles even though the model has access to the previously generated audio tokens.

These findings suggest that the speech model learns to primarily focus on the small span of corresponding text hidden states without heavily modeling the semantic space of the entire text and

Table 1: Comparison of recent speech models, VITA-Audio leverages the hidden state to enhance model performance, adopts an end-to-end architecture, and achieves zero audio token delay.

| Model | | Audio Token Delay | Leveraging Hidden States | End-to-End |
|---|---|---|---|---|
| Freeze-Omni | [56] | Text Length | ✓ | ✗ |
| MinMo | [9] | 5 | ✓ | ✗ |
| Mini-Omni | [58] | 7 | ✗ | ✓ |
| Moshi | [19] | 1 | ✗ | ✓ |
| GLM-4-Voice | [66] | 13 | ✗ | ✓ |
| LUCY | [28] | 7 | ✗ | ✓ |
| VITA-Audio | | 0 | ✓ | ✓ |

audio sequence. This discovery instills confidence that we can learn the simple mapping relationship between text hidden states and audio tokens with relatively simple modules and without relying on the extensive semantic modeling of LLMs.

In this paper, we introduce VITA-Audio, a lightweight framework that uses separate efficient modules, named Multiple Cross-modal Token Prediction (MCTP), to efficiently generate audio responses from text embeddings and LLM hidden states. This approach enables obtaining both text tokens and an audio chunk in a single LLM forward pass, achieving zero delay in audio tokens. A comparison of the delay of the first audio token is presented in Table 1 , where we define "audio token delay" as the number of additional LLM forward steps required to generate the first audio token after the first LLM forward pass. We distinguish this delay from "audio generation delay" which is the number of additional LLM forward passes to generate a meaningful and consistent chunk of audio. VITA-Audio has both zero audio token delay and zero audio generation delay.

To this end, through a four-stage progressive training strategy, we construct a set of lightweight yet powerful MCTP modules, which predict **10** audio tokens directly from historical inputs and LLM hidden states without requiring additional LLM forward passes, thus significantly enhancing the model's inference speed without sacrificing audio quality.

In summary, our main contributions are as follows.

- We introduce VITA-Audio, the first end-to-end speech model capable of generating audio during the first forward pass. Leveraging audio generation without relying on extensive text semantic modeling capabilities, VITA-Audio designs lightweight MCTP modules to generate decodable audio token chunks with zero audio token delay, thus overcoming the real-time limitations in traditional cascaded models and existing end-to-end methods.

- VITA-Audio achieves remarkable end-to-end inference acceleration by generating ten audio tokens in a single forward pass, resulting in $3 \sim 5\times$ speedup when implemented on a 7B LLM while preserving the ability of high-quality speech synthesis.

- We fully release VITA-Audio to the open-source community. Although VITA-Audio is trained on open-source data only, comprehensive evaluations reveal that VITA-Audio achieves the state-of-the-art performance on multiple benchmarks for ASR, TTS, and SQA tasks, outperforming existing models in both efficiency and accuracy, especially the open-source ones with a similar parameter scale, therefore setting a new standard for real-time speech-to-speech models.

## 2   Related Work

Large language models (LLMs) have revolutionized human-computer interaction with advanced natural language processing. Extending these capabilities to speech—a natural communication modality—has become a key research focus. Traditional speech interaction systems [54, 32, 69] typically adopt a cascaded architecture, combining separate ASR, LLM, and TTS modules. However, this approach suffers from increased latency, loss of paralinguistic cues, and error propagation.

Recent work [24, 49] has improved integration by connecting audio encoders to LLMs via trainable adapters, but still relies on independent TTS modules. To address this, some methods incorporate

LLM hidden states into audio decoders. For example, Llama-Omni [23] uses a non-autoregressive transformer to predict audio tokens from upsampled LLM states, while Freeze-omni [56] freezes the LLM and combines autoregressive and non-autoregressive decoders. Minmo [9] integrates a language model with CosyVoice2 [22] for mixed speech-text processing.

End-to-end models further unify TTS within LLMs, enabling direct text and speech generation. These models follow either parallel or interleaved audio-text modeling. In the parallel modeling paradigm, the model uses different heads to process hidden states, generating both text and multiple audio tokens [10, 19]. Since the input to the LLM is altered during autoregression, maintaining the original capabilities of the LLM presents significant challenges. To perform inference without large-scale pretraining, Mini-Omni [58] and LUCY [28] rely on batch parallel decoding to preserve the inference capability of the LLM.

Compared to parallel-paradigm models, interleave-paradigm models appear to better preserve language capabilities, as suggested by the performance comparison on spoken question-answering benchmarks [66]. We attribute this difference to the fact that parallel-paradigm models use an average of text and audio representations as input, which significantly diverges from the inputs used during pretraining. However, interleave-paradigm models face a latency issue due to their sequential prediction of audio tokens, especially when the audio token rate is high.

VITA-Audio leverages the strengths of these architectures by adopting the interleaved modeling paradigm and introducing MCTP for audio generation. The former maximally preserves the LLM's language ability, and the latter reduces inference latency by generating multiple audio tokens in a single forward pass.

# 3 Method

## 3.1 Overview

As illustrated in Fig. 2, VITA-Audio consists of four major components: an audio encoder, an audio decoder, a large language model backbone, and a set of Cross-modal Token Prediction (MCTP) modules. The audio signal is first processed by the audio encoder, whose output is then fed into the LLM for further processing. During each forward pass, the LLM alternately generates text and audio tokens. The hidden states from the final layer of LLM, along with the embedding of the predicted token, are provided as input to the MCTP modules. The historical input tokens, the tokens predicted by the LLM, and by the MCTP modules are concatenated to form the inputs to the next LLM forward pass. Finally, the audio tokens generated by both the LLM and the MCTP modules are aggregated and passed to the audio decoder to generate the final audio output.

## 3.2 Multiple Cross-Modal Token Prediction (MCTP) Module

As described in Section 1, the text and speech modalities exhibit a monotonic alignment pattern. This cross-modal alignment allows us to avoid complex modeling of the semantic latent space and to focus on learning a simple text-to-speech mapping relationship, which we propose to use lightweight modules to learn. In the preliminary experiments, we use a few lightweight Transformer blocks to predict multiple audio tokens from LLM hidden states, and embed the predicted tokens into the LLM's autoregressive inference.

Standard autoregressive modeling can be formulated as:

$$p_t(Y_{t-1}, ..., Y_0) \equiv P[Y_t|Y_{t-1}, ..., Y_0], \tag{1}$$

where $Y_t$ denotes the predicted audio token at time step $t$, and $p_t$ represents the conditional probability distribution based on the historical sequence. When extended to multi-step prediction, *i.e.*, predicting the $i$-$th$ audio token at time step $t + i$, the formulation becomes as:

$$p_{t+i}(Y_{t-1}, ..., Y_0) \equiv \widetilde{P}[Y_{t+i}|Y_{t-1}, ..., Y_0]. \tag{2}$$

At this point, there is a significant deviation in the consistency of the distribution between $\widetilde{P}$ and $P$. As $i$ increases, the difference between the two distributions will progressively widen, resulting in a growing accumulation of errors and leading to poor mapping between text and audio.

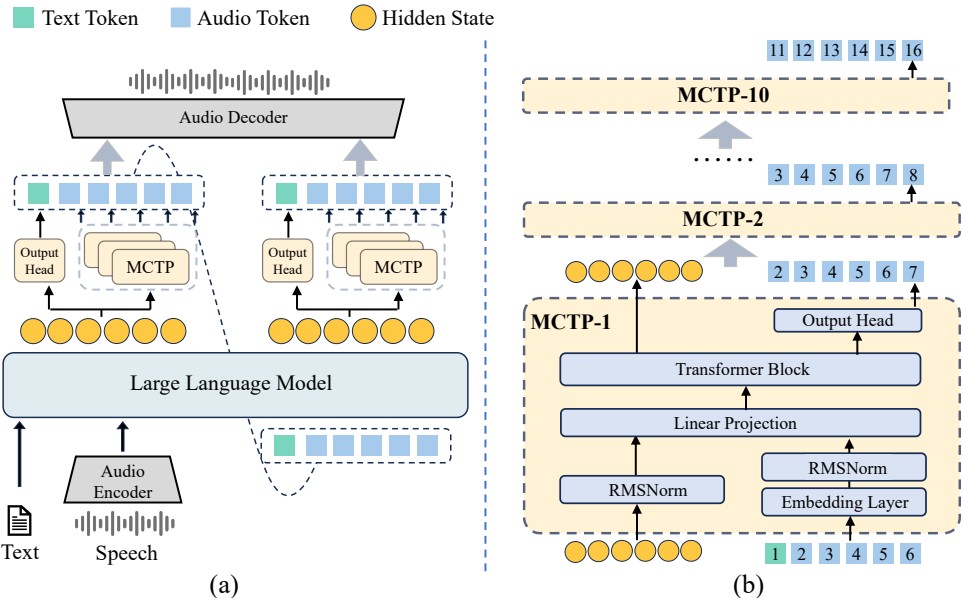

Figure 2: Architecture overview. (a) VITA-Audio is an end-to-end large speech model equipped with 10 light-weight Multiple Cross-modal Token Prediction (MCTP) modules that enable speech generation with extremely low latency. As shown in Fig. 1, we observe that the hidden states of certain text tokens in the LLM backbone contain sufficient semantic information for generating the corresponding audio tokens, which means that it is unnecessary to attend to additional text tokens when generating audio. Thus, we propose to utilize a set of light-weight MCTP modules to model the mapping from LLM hidden states to the audio tokens. (b) The details of the MCTP modules. Our MCTP module has a light-weight architecture, which enables it to finish one forward pass within 0.0024 seconds ( 11% of the LLM backbone). The MCTP module is capable of generating 10 audio tokens from the LLM hidden states and the text embedding, and the generated audio tokens can be decoded by the audio decoder directly. The utilization of MCTP modules enables VITA-Audio to generate audio responses in one LLM forward pass, which achieves extremely fast generation speed.

To address this issue, we adopt a cascaded prediction architecture. Specifically, the hidden states and output sequence from the preceding modules are employed as joint input conditions for the subsequent modules:

$$p_{t+i}(Y_{t-1}, \ldots, Y_0) \equiv \widetilde{P}[Y_{t+i} | Y_{t-1}, \ldots, Y_0, h_{t+i-1}, o_{t+i-1}, \ldots, o_t], \tag{3}$$

where $h_{t+i-1}$ and $o_{t+i-1}$ represent the hidden state and output sequence of the preceding module, respectively. By introducing progressively updated contextual information, modules can achieve incremental optimization of cross-modal mapping, ensuring accurate modality synchronization at each time step.

Inspired by DeepSeek V3 [18], we adopt an isomorphic Multi-Token Prediction (MTP) framework to construct our MCTP module. Unlike DeepSeekV3 and speculative decoding [38], where the former focuses on improving training and the latter requires verification to ensure that the sampling distribution matches exactly with that of the original model, we use the MCTP module for audio-text mapping, presumably a simpler task than semantic modeling. As a result, we require a comparatively small amount of text data to train our model.

Since the embedding layer and output heads are shared with the LLM, the audio tokens generated by the MCTP module are directly incorporated into the autoregressive process of the LLM. As illustrated in Fig. 2, the hidden states and output token, from the LLM or the preceding MCTP module, are concatenated with the input tokens and fed into a Transformer block for next-step processing. The resulting hidden states and token are then passed to the next MCTP module. Upon completion of a forward pass, the audio tokens generated by either the LLM or the MCTP modules are aggregated as the input sequence for the subsequent LLM forward iteration.

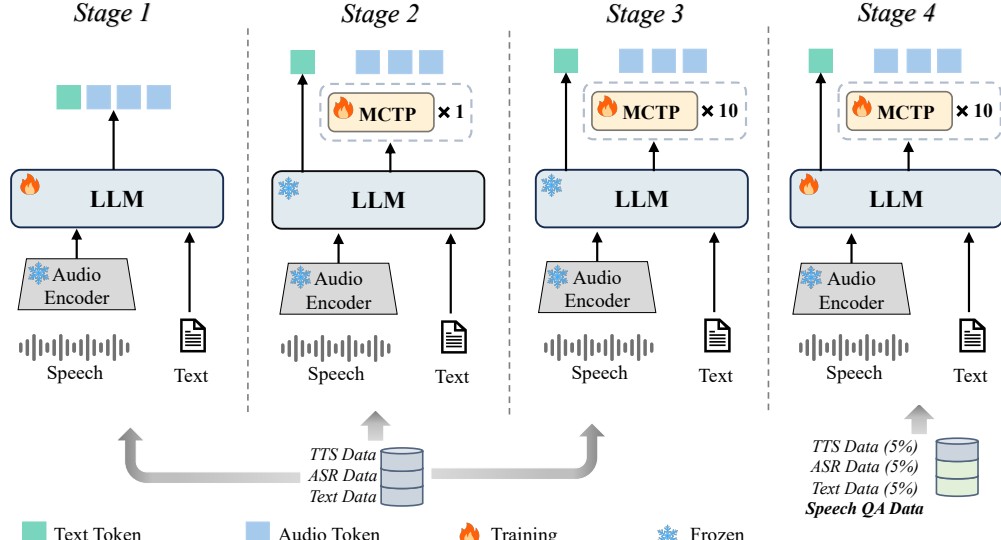

Figure 3: Training pipeline of VITA-Audio. The first stage (Audio-Text Alignment) enhances the LLM by extending its audio modeling capability through large-scale speech pre-training. The second stage (Single MCTP module Training) connects an MCTP module with the LLM to predict one subsequent token based on the input tokens and the LLM's hidden states. The third stage (Multiple MCTP Modules Training) increases the number of MCTP modules in the model to predict more tokens in each model forward. The last stage (Supervised Fine-tuning) provides the speech-to-speech capability to the model by optimizing it on the large-scale speech QA dataset.

## 3.3 Training

### 3.3.1 Data Construction

VITA-Audio is trained exclusively on open-source datasets, integrating multi-domain and multi-language speech data resources. The training dataset encompasses a diverse range of sources. Detailed descriptions of the datasets used at each stage are provided in Table E1 in the Appendix.

All training data are uniformly packed into sequences of fixed length (8K tokens), an approach that enables effective training on samples of varying lengths [47]. We reinitialize the positional embeddings and attention masks for all packed samples to ensure that the model attends exclusively to tokens within the same original sample. This processing strategy not only eliminates potential artifacts introduced by data concatenation but also significantly enhances training stability and reduces computational overhead.

### 3.3.2 Training Pipline

For VITA-Audio to output a consistent sequence of audio tokens in a single forward pass, each MCTP module must model a distinct distribution. As a result, training all the MCTP modules simultaneously becomes a challenging task, especially when the number of modules is large, due to potentially misaligned optimization objectives. We propose a four-stage training strategy, as shown in Fig. 3, to progressively equip the MCTP modules with the ability to map text to its audio, thereby reducing the difficulty of their convergence. Further training details are provided in Sec. B of the Appendix.

## 3.4 Inference

In order to address diverse scenarios, four distinct inference paradigms have been designed as shown in Fig. 4.

For the ASR and TTS tasks, we propose VITA-Audio-Turbo. In each forward pass, the LLM generates one token, followed by the generation of ten tokens by the MCTP modules. This paradigm

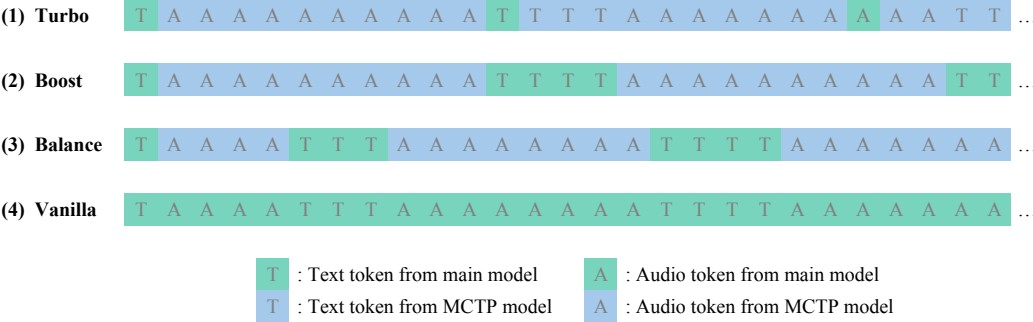

Figure 4: The four text-audio interleaved inference modes are illustrated as follows: 1) Turbo: As the fastest inference mode, it generates 1 token by the main model and 10 additional tokens via MCTP in each forward pass. To ensure that a valid audio chunk is decoded after the first forward pass, the first generated 11 tokens are split into 1 text token and 10 audio tokens. Then, the Turbo mode iteratively generates 4 text tokens and 10 audio tokens in the following forward. 2) Boost: To enhance the quality of text tokens, Boost mode follows the text-audio cyclic pattern of Turbo mode, with the main model generating every text token and MCTP generating every audio token. 3) Balance: To keep a balanced text-audio ratio, $i.e.$, $1 : 2$, the balance mode further changes the text-audio cyclic pattern of the Boost mode. Specifically, the balance mode sequentially generates 1 text token from the main model, 4 audio tokens (2 tag tokens mark the beginning and end of audios, and 2 common tokens denote the audio content) from MCTP, 3 text tokens from the main model, 8 text tokens (2 tag tokens mark the beginning and end of audios, and 6 common tokens denote the audio content) from MCTP, and then iteratively generates 4 text tokens from the main model and 10 audio tokens (2 tag tokens mark the beginning and end of audios, and 8 common tokens denote the audio content) from MCTP. 4) Vanilla: As the slowest inference mode, Vanilla mode follows the text-audio cyclic pattern of Balance mode, with the main model generating every token.

is the most efficient among the options; however, the performance of speech dialogue tasks degrades due to the need to predict the text token.

For speech dialogue tasks, we introduce VITA-Audio-Boost and VITA-Audio-Balance. Their main difference lies in audio token generation: VITA-Audio-Boost generates eight directly decodable audio tokens in the first forward pass, whereas VITA-Audio-Balance adheres to a strict $1 : 2$ text-to-audio token ratio for enhanced speech quality. The latter requires two forward passes to generate enough audio tokens for decoding. To optimize model performance, distinct models were trained for each of the two modes.

VITA-Audio-Vanilla is designed for scenarios that require higher language performance. It generates tokens solely using the main LLM, sacrificing efficiency but offering a slight performance boost.

## 4  Experiment

### 4.1  Experiment Settings

We use the Qwen2.5-7B-Instruct [45] as the pre-trained text LLM. The initial version of VITA-Audio utilizes the speech tokenizer and speech decoder in GLM-4-Voice [66], which effectively captures semantic information at an ultra-low bitrate. In the second version, $i.e.$, VITA-Audio-Plus further replaces the GLM-4-Voice tokenizer with SenseVoiceSmall [1] and an MLP-based adapter. The detail comparison between VITA-Audio and VITA-Audio-Plus is listed in Table E2.

### 4.2  Evaluation on Spoken Question Answering

We evaluate the spoken question answering capability of VITA-Audio on three public English datasets: Web-Questions [5], Llama-Question [42], and TriviaQA [33]. Two evaluation methods are employed: S→T, where the text responses generated by the model are evaluated directly, and S→S, where the model's speech responses are transcribed using Whisper [46] before evaluation.

Table 2: Results on Spoken Question Answering (SQA) benchmarks. "S$x$" denotes the $x$-th training stage of speech models.

| Model | | #Params | Llama Question | | TriviaQA | | Web Question | | Mean | |
|---|---|---|---|---|---|---|---|---|---|---|
| | | | S→T | S→S | S→T | S→S | S→T | S→S | S→T | S→S |
| Proprietary Models | | | | | | | | | | |
| MinMo | [9] | 7B | 78.9 | 64.1 | 48.3 | 37.5 | 55.0 | 39.9 | 60.7 | 47.2 |
| Open-source Models | | | | | | | | | | |
| Moshi | [19] | 7B | 62.3 | 21.0 | 22.8 | 7.3 | 26.6 | 9.2 | 37.2 | 12.5 |
| GLM-4-Voice | [66] | 9B | 64.7 | 50.7 | 39.1 | 26.5 | **55.0** | 39.9 | 45.3 | 31.0 |
| LUCY (S2) | [28] | 7B | 59.6 | 51.0 | 23.2 | 18.2 | 26.6 | 18.2 | 36.5 | 29.1 |
| VITA-Audio-Boost | | 7B | 68.7 | 60.3 | 30.5 | 29.3 | 32.9 | 30.4 | 44.0 | 40.0 |
| VITA-Audio-Vanilla | | 7B | 71.3 | 66.3 | 31.9 | 30.1 | 33.5 | 31.4 | 45.6 | 42.6 |
| VITA-Audio-Plus-Boost | | 7B | **76.3** | 64.6 | 43.6 | 39.5 | 44.2 | 40.0 | 54.7 | 48.0 |
| VITA-Audio-Plus-Vanilla | | 7B | 75.6 | **68.0** | **45.9** | **42.7** | 45.0 | **41.7** | **55.5** | **50.8** |

Table 3: Results on Text to Speech (TTS) Benchmarks. "S$x$" denotes the $x$-th training stage.

| Model | | Seed-TTS | | | LibriTTS |
|---|---|---|---|---|---|
| | | *test-zh* | *test-en* | *test-hard* | *test-clean* |
| | | CER (%) ↓ | WER (%) ↓ | WER (%) ↓ | WER (%) ↓ |
| Seed-TTS | [2] | 1.12 | 2.25 | 7.59 | – |
| CosyVoice | [21] | 3.63 | 4.29 | 11.75 | 2.89 |
| CosyVoice2 | [22] | 1.45 | 2.57 | **6.83** | 2.47 |
| VITA-1.5 (S3) | [25] | 8.44 | 2.63 | – | – |
| GLM-4-Voice | [66] | 2.91 | 2.10 | – | 5.64 |
| VITA-Audio-Turbo (S1) | | 1.18 | 1.92 | 10.58 | 1.96 |
| VITA-Audio-Turbo (S2) | | **0.96** | 1.92 | 9.72 | 1.98 |
| VITA-Audio-Turbo (S3) | | 1.05 | **1.77** | 9.86 | 1.99 |
| VITA-Audio-Turbo (S4) | | 1.07 | 2.26 | 10.08 | 2.08 |
| VITA-Audio-Plus-Boost | | 1.32 | 2.21 | 12.05 | 2.21 |
| VITA-Audio-Plus-Vanilla | | 1.13 | 1.85 | 10.21 | **1.89** |

We compare VITA-Audio with the latest speech models that have comparable parameter sizes, and the results are shown in Table 2. Our model demonstrates superior performance in the S→T task and achieves SOTA results in the S→S setup. It is particularly noteworthy that the training approach of VITA-Audio ensures minimal degradation between S→T and S→S, with a performance drop of only 9%. This indicates that VITA-Audio achieves high-quality alignment between text and speech modalities, with benefits extending beyond processing speed alone.

## 4.3 Evaluation on Fundamental Speech Competence

**TTS**  We evaluate the TTS performance of VITA-Audio on Seed-TTS [2] and LibriTTS [65] benchmarks. We use Whisper-Large-V3 [46] and Paraformer[29] to transcribe into text the generated English and Chinese speech, respectively.

We present the results of VITA-Audio at each stage in Table 3. In these results, the output of VITA-Audio's first stage (S1) consists of text tokens directly generated by the LLM, while the outputs of the second (S2), third (S3), and fourth (S4) stages are alternately generated by both the LLM and the MCTP module. The experiments demonstrate that VITA-Audio outperforms other open-source models with a similar number of parameters. Additionally, it should be noted that, despite using ten MCTP modules for accelerated inference, VITA-Audio's TTS capabilities are largely preserved throughout the training process, further validating the effectiveness of the MCTP module in aligning text and audio.

**ASR**  We evaluate the ASR performance of the four stages of VITA-Audio on WenetSpeech [67], AIshell [6], LibriSpeech [43], and Fleurs [15], and a subset of the results are reported in Table 4.

Table 4: Results on Automatic Speech Recognition (ASR) Benchmarks. "S$x$" denotes the $x$-th training stage. Compared to other methods, **VITA-Audio is trained with open-source data only**.

| Model | | WenetSpeech | | AIShell | LibriSpeech | |
|---|---|---|---|---|---|---|
| | | test_meeting ↓ | test_net ↓ | test ↓ | test-clean ↓ | test-other ↓ |
| Qwen2-Audio-base | [12] | 8.40 | 7.64 | 1.52 | **1.74** | **4.04** |
| Baichuan-Audio-base | [37] | 13.28 | 10.13 | 1.93 | 3.02 | 6.04 |
| VITA-1.5 (S3) | [24] | 10.0 | 8.4 | 2.2 | 3.4 | 7.5 |
| Freeze-Omni | [56] | 13.46 | 11.8 | 2.48 | 3.82 | 9.79 |
| LUCY (S1) | [28] | 10.42 | 8.78 | 2.40 | 3.36 | 8.05 |
| Step-Audio-chat | [49] | 10.83 | 9.47 | 2.14 | 3.19 | 10.67 |
| Qwen2.5-Omni | [59] | 7.71 | **6.04** | **1.13** | 2.37 | 4.21 |
| VITA-Audio-Vanilla | | 17.34 | 13.45 | 4.46 | 2.98 | 8.07 |
| VITA-Audio-Plus-Boost | | 9.38 | 8.97 | 4.72 | 3.13 | 7.07 |
| VITA-Audio-Plus-Vanilla(S1) | | **6.68** | 6.59 | 1.51 | 1.91 | 4.29 |
| VITA-Audio-Plus-Vanilla | | 7.12 | 6.90 | 1.94 | 2.00 | 4.60 |

Table 5: Boostup Ratio under Different Inference Paradigms.

| Mode | Model Size | #GPU | Total Second ↓ | Token Per Second ↓ | Speedup ↑ |
|---|---|---|---|---|---|
| Vanilla | | | 53.89 | 76.00 | 1.00 × |
| Boost | 0.5B | 1 | 20.65 | 198.35 | 2.61 × |
| Balance | | | 20.71 | 197.78 | 2.60 × |
| Turbo | | | 11.83 | 346.24 | 4.56 × |
| Vanilla | | | 63.38 | 64.62 | 1.00 × |
| Boost | 7B | 1 | 23.97 | 170.88 | 2.64 × |
| Balance | | | 23.94 | 171.09 | 2.64 × |
| Turbo | | | 13.43 | 304.99 | 4.72 × |
| Vanilla | | | 255.13 | 16.05 | 1.00 × |
| Boost | 72B | 2 | 84.98 | 48.20 | 3.00 × |
| Balance | | | 85.13 | 48.11 | 3.00 × |
| Turbo | | | 39.5 | 103.60 | 6.46 × |

More detailed results can be found in Table E3 and Table E4 in the Appendix. The results for other works are partially reproduced from their respective original works for comparison.

It can be observed that VITA-Audio-plus-vanilla demonstrates highly competitive performance across various benchmarks. Moreover, VITA-Audio-plus-Boost achieves remarkably fast inference speed while still maintaining strong overall performance.

## 4.4 Evaluation of Latency

**Inference Speedup** Efficient mapping between text and speech is the core of VITA-Audio. To demonstrate its effectiveness, we compare the inference time across different modes of VITA-Audio for various model sizes. Specifically, we evaluate the inference speed on GPUs capable of 148 TFLOPS under bfloat16 precision, with the output fixed at 4096 tokens, and record the total time as the model's inference time. All models, regardless of size, are randomly initialized, and this initialization do not affect the inference time measurements. We use Transformers [57] and FlashAttention-2 [17].

As mentioned in Section 3.4, VITA-Audio-Vanilla only uses the main model for output; VITA-Audio-Turbo uses both the main model and all the MCTP modules during each forward pass; and VITA-Audio-Boost and VITA-Audio-Balance progressively increase the number of MCTP modules used to ensure higher accuracy.

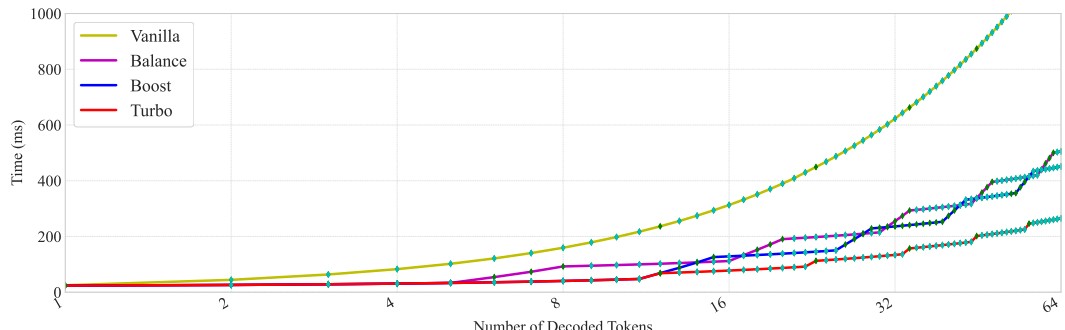

Figure 5: Token generation speed curves of four text-audio interleaved modes.

As shown in Table 5, we present a comparison of the time consumption for different inference modes and model sizes. Noted that the time consumption does not include the cost of the audio encoder and audio decoder. We observe that in VITA-Audio-Turbo, a speedup of approximately $5\times$ is achieved across models ranging from 0.5B to 72B, greatly enhancing the output token throughput. VITA-Audio-Boost also achieves around $3\times$ acceleration across various sizes, resulting in a desirable performance for real-time speech dialogue systems. For example, 72 B VITA-Audio generates approximately 40 tokens per second, which, excluding generated text tokens, corresponds to roughly three seconds of audio and associated text when using a 12.5Hz speech tokenizer. This performance is sufficiently fast for human-computer interaction.

**Latency** In human-computer interaction, latency is a crucial metric, as it determines whether users can interact with the model in real-time. Given that most speech models support streaming output, the key to reducing perceived latency lies in shortening the time required to generate the first chunk of audio.

We visualize the timeline of model decoding phrase in Fig. 5. The green marks denote the tokens generated by the main model, and the blue marks are the tokens generated by MCTP modules. We set the number of prefiil tokens to 32. And Fig. 5 shows that VITA-Audio-Turbo completes the generation of the first audio chunk in about 50 ms, while VITA-Audio-Vanilla requires about 220 ms. VITA-Audio-Boost and VITA-Audio-Balance generate fewer audio tokens in the first forward and more text audio tokens in the following forward. Thus, they are slower than VITA-Audio-Turbo but still significantly faster than VITA-Audio-Vanilla.

Thanks to the advantage of zero audio generation delay, VITA-Audio produces multiple audio tokens in the first forward pass, allowing the first audio token chunk to be generated during the initial forward pass, which can then be used for decoding. This significantly reduces the perceived delay. In the experimental environment previously mentioned, VITA-Audio reduces the time to generate the first audio token chunk from 236 to 53 ms, as shown in Table E7.

## 5 Conclusion

In this paper, we introduce VITA-Audio, a lightweight framework that uses separate efficient modules, named Multiple Cross-modal Token Prediction (MCTP) modules, to efficiently generate audio responses from text embeddings and LLM hidden states. MCTP learns the simple mapping relationship between text hidden states and audio tokens with relatively simple modules and without relying on the extensive semantic modeling of LLMs. Our model achieves new state-of-the-art performance on multiple benchmarks for ASR, TTS, and SQA tasks, outperforming existing models in efficiency and accuracy, especially the open-source ones with a similar parameter scale. Therefore, it sets a new standard for real-time speech-to-speech models.

## Acknowledgments

This work is partially funded by National Natural Science Foundation of China (Grant No. 62506158 and No. 62441234), and CCF-Tencent Rhino-Bird Open Research Fund.

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

# VITA-Audio: Fast Interleaved Cross-Modal Token Generation for Efficient Large Speech-Language Model

## Supplementary Material

## A  Training Data

**ASR Data**  We aggregated approximately $100,000$ hours of open-source Automatic Speech Recognition (ASR) data, including WenetSpeech [67], Librispeech [43], Multilingual LibriSpeech [44], Common Voice 17 [43], MMCRSC [40], GigaSpeech [8], People's Speech [27], VoxPopuli [53], and the AISHELL series (AISHELL-1 [6] to AISHELL-4 [26]).

**TTS Data**  Concurrently, we integrate approximately $100,000$ hours of open-source Text-to-Speech (TTS) data, primarily consisting of the Wenetspeech4TTS [39], LibriTTS [65], GLOBE [55], and Emilia [30] datasets.

**Speech QA Data**  For speech question-answering (Speech-QA), we utilize VoiceAssistant-400K [58] and AudioQA-1.0M [28], totaling $1.4$ million speech QA data, to enhance the model's speech-to-speech dialogue capabilities.

**Text-Only Data**  The pure text data is collected from OpenHermes-2.5 [50] LIMA [70], databricks-dolly-15k [16], MetaMathQA [62], MathInstruct [64], Orca-Math [41], atlas-math-sets [52], goat [51], and camel-ai-math [36]. Given that discrete audio token sequences exhibit significantly longer lengths compared to their textual counterparts, we incorporate several specialized long-context text datasets following the Long-VITA [47] to enhance contextual modeling capabilities. These include Long-Instruction [63], LongForm [35], LongAlign-10k [3], LongCite-45k [68], LongWriter-6k [4], LongQLoRA [60], LongAlpaca [11], and LongData-Collections [14].

## B  Training Pipline

**Stage 1: Audio-Text Alignment.** Building upon pretrained language models, the goal of this stage is to extend the audio modeling capabilities to the LLM through large-scale speech pretraining. We freeze the audio encoder and audio decoder, and train the LLM using ASR, TTS, and Text-only data. During this stage, the output of the LLM can be either pure text tokens or audio tokens.

**Stage 2: Single MCTP Module Training.** After Stage 1, the model learns both text and audio distributions. The objective of Stage 2 is to train the initial MCTP module to predict one subsequent token based on the output tokens and hidden states from the LLM. This stage employs the same dataset configuration as Stage 1. We initialize the MCTP module using parameters from the final layer of the LLM, with the gradient detached from the LLM.

**Stage 3: Multiple MCTP Modules Training.** The objective of this stage is to extend the single MCTP module to multiple MCTP modules. Specifically, each MCTP module predicts the token at its corresponding position given the output tokens and hidden states of the previous MCTP module. All subsequent MCTP modules are initialized using the weights of the MTCP module from Stage 2. This stage also incorporates gradient detachment to optimize the model training process.

**Stage 4: Supervised Fine-tuning.** After the previous three training stages, VITA-Audio has acquired the ability to efficiently and accurately map text to audio. To enable speech-to-speech dialogue capability, we then conduct supervised fine-tuning using speech QA datasets while maintaining a small amount of TTS, ASR, and text-only data to ensure training stability. To optimize training effectiveness, different learning rates are used for the MCTP module and the main LLM. For the speech-to-speech data, we employ an interleaved output format. This design enforces the model to initiate audio token generation during the first forward pass, enabling synchronized decoding of the audio tokens rather than waiting until all text tokens have been generated.

# C   Limitations

While our approach enables efficient generation of audio tokens, the overall end-to-end latency remains above the theoretical lower bound, primarily due to the constrained generation speed of the audio decoder. Further improving the response speed of the audio decoder in end-to-end speech models is a worthwhile direction for future exploration.

# D   Data Format

**Speech QA Interleaved Data Format**

```
{
    "messages": [
    {
        "role": "user",
        "content": "<|begin_of_audio|> audio_sequence_1 <|end_of_audio|>"
    },
    {
        "role": "assistant",
        "content": "text_sequence_1 <|begin_of_audio|>
        audio_sequence_2 <|end_of_audio|> text_sequence_2
        <|begin_of_audio|> audio_sequence_3 <|end_of_audio|>"
    },
    {
        "role": "user",
        "content": "<|begin_of_audio|> audio_sequence_4 <|end_of_audio|>"
    },
    {
        "role": "assistant",
        "content": "text_sequence_3 <|begin_of_audio|>
        audio_sequence_5 <|end_of_audio|> text_sequence_4
        <|begin_of_audio|> audio_sequence_6 <|end_of_audio|>"
    }
    ]
}
```

**Prompt for TTS task.**

```
{
    "messages": [
    {
        "role": "user",
        "content": "Convert the text to speech.\ntext_sequence"
    }
    ]
}
```

# E   Figures and Tables

Table E1: Summary of datasets used in VITA-Audio for different stages.

| Task | Name | | Total Number | Sampling Ratio | | | |
|------|------|--|-------------|---------|---------|---------|---------|
| | | | | Stage 1 | Stage 2 | Stage 3 | Stage 4 |
| ASR | WenetSpeech | [67] | 10,000H | 1.0 | 1.0 | 1.0 | 0.05 |
| | Librispeech | [43] | 1,000H | 1.0 | 1.0 | 1.0 | 0.05 |
| | Multilingual LibriSpeech | [44] | 71,506H | 1.0 | 1.0 | 1.0 | 0.05 |
| | Common Voice 17 | [43] | 2.849H | 1.0 | 1.0 | 1.0 | 0.05 |
| | MMCRSC | [40] | 755H | 1.0 | 1.0 | 1.0 | 0.05 |
| | GigaSpeech | [8] | 10,000H | 1.0 | 1.0 | 1.0 | 0.05 |
| | People's Speech | [27] | 1,000H | 1.0 | 1.0 | 1.0 | 0.05 |
| | VoxPopuli | [53] | 543H | 1.0 | 1.0 | 1.0 | 0.05 |
| | AISHELL-1 | [6] | 170H | 1.0 | 1.0 | 1.0 | 0.05 |
| | AISHELL-2 | [20] | 1,000H | 1.0 | 1.0 | 1.0 | 0.05 |
| | AISHELL-3 | [48] | 85H | 1.0 | 1.0 | 1.0 | 0.05 |
| | AISHELL-4 | [26] | 120H | 1.0 | 1.0 | 1.0 | 0.05 |
| TTS | Wenetspeech4TTS | [39] | 12,800H | 1.0 | 1.0 | 1.0 | 0.05 |
| | LibriTTS | [65] | 585H | 1.0 | 1.0 | 1.0 | 0.05 |
| | GLOBE | [55] | 535H | 1.0 | 1.0 | 1.0 | 0.05 |
| | Emilia | [30] | 96,700H | 1.0 | 1.0 | 1.0 | 0.05 |
| Speech QA | VoiceAssistant-400K | [58] | 400K | 0.0 | 0.0 | 0.0 | 2.0 |
| | AudioQA-1.0M | [28] | 1M | 0.0 | 0.0 | 0.0 | 2.0 |
| Text QA | OpenHermes-2.5 | [50] | 1M | 1.0 | 1.0 | 1.0 | 0.05 |
| | LIMA | [70] | 1K | 1.0 | 1.0 | 1.0 | 0.05 |
| | databricks-dolly-15k | [16] | 15K | 1.0 | 1.0 | 1.0 | 0.05 |
| | MetaMathQA | [62] | 395K | 1.0 | 1.0 | 1.0 | 0.05 |
| | MathInstruct | [64] | 262K | 1.0 | 1.0 | 1.0 | 0.05 |
| | Orca-Math | [41] | 200K | 1.0 | 1.0 | 1.0 | 0.05 |
| | atlas-math-sets | [52] | 17.8M | 1.0 | 1.0 | 1.0 | 0.05 |
| | goat | [51] | 1.7M | 1.0 | 1.0 | 1.0 | 0.05 |
| | camel-ai-math | [36] | 50K | 1.0 | 1.0 | 1.0 | 0.05 |
| Long Text QA | Long-Instruction | [63] | 16K | 1.0 | 1.0 | 1.0 | 0.05 |
| | LongForm | [35] | 23K | 1.0 | 1.0 | 1.0 | 0.05 |
| | LongAlign-10k | [3] | 10K | 1.0 | 1.0 | 1.0 | 0.05 |
| | LongCite-45k | [68] | 45K | 1.0 | 1.0 | 1.0 | 0.05 |
| | LongWriter-6k | [4] | 6K | 1.0 | 1.0 | 1.0 | 0.05 |
| | LongQLoRA | [60] | 39K | 1.0 | 1.0 | 1.0 | 0.05 |
| | LongAlpaca | [11] | 12K | 1.0 | 1.0 | 1.0 | 0.05 |
| | Long-Data-Collections | [14] | 98K | 1.0 | 1.0 | 1.0 | 0.05 |

Table E2: Comparison of model structures between VITA-Audio and VITA-Audio-Plus.

| Name | Base LLM | Audio Encoder | Audio Adapter | Audio Decoder |
|---|---|---|---|---|
| VITA-Audio | Qwen2.5-7B [45] | GLM-4-Voice-Tokenizer [66] | – | GLM-4-Voice-Decoder [66] |
| VITA-Audio-Plus | Qwen2.5-7B [45] | SenseVoiceSmall [1] | MLP | GLM-4-Voice-Decoder [66] |

Table E3: Results on Automatic Speech Recognition (ASR) Benchmarks. "S$x$" denotes the $x$-th training stage. Compared to other methods, **VITA-Audio is trained with open-source data only**.

| Datasets | Model | | WER (%) ↓ |
|---|---|---|---|
| LibriSpeech [43] *test-clean \| test-other* | Qwen2-Audio-base | [12] | **1.74** \| **4.04** |
| | Baichuan-Audio-base | [37] | 3.02 \| 6.04 |
| | Freeze-Omni | [56] | 3.82 \| 9.79 |
| | VITA-1.5 (S3) | [25] | 3.40 \| 7.50 |
| | LUCY (S1) | [28] | 3.36 \| 8.05 |
| | Step-Audio-chat | [49] | 3.19 \| 10.67 |
| | Qwen2.5-Omni | [59] | 2.37 \| 4.21 |
| | VITA-Audio-Turbo | | 6.29 \| 12.86 |
| | VITA-Audio-Vanilla | | 2.98 \| 8.07 |
| | VITA-Audio-Plus-Boost | | 3.13 \| 7.07 |
| | VITA-Audio-Plus-Vanilla (S1) | | 1.91 \| 4.29 |
| | VITA-Audio-Plus-Vanilla | | 2.00 \| 4.60 |
| Fleurs [15] *zh \| en* | Qwen2-Audio-base | [12] | 3.63 \| 5.20 |
| | Baichuan-Audio-base | [37] | 4.15 \| 8.07 |
| | Step-Audio-chat | [49] | 4.26 \| 8.56 |
| | Qwen2.5-Omni | [59] | **2.92** \| **4.17** |
| | VITA-Audio-Plus-Vanilla | | 3.69 \| 4.54 |

Table E4: Results on Automatic Speech Recognition (ASR) Benchmarks. "S$x$" denotes the $x$-th training stage. Compared to other methods, **VITA-Audio is trained with open-source data only**.

| Datasets | Model | | WER (%) ↓ |
|---|---|---|---|
| | Qwen2-Audio-base | [12] | 1.52 |
| | Baichuan-Audio-base | [37] | 1.93 |
| | Freeze-Omni | [56] | 2.48 |
| | LUCY (S1) | [28] | 2.40 |
| | Step-Audio-chat | [49] | 2.14 |
| AISHELL-1 [6] | Qwen2.5-Omni | [59] | **1.13** |
| | VITA-Audio-Turbo | | 7.70 |
| | VITA-Audio-Vanilla | | 4.46 |
| | VITA-Audio-Plus-Boost | | 4.72 |
| | VITA-Audio-Plus-Vanilla (S1) | | 1.51 |
| | VITA-Audio-Plus-Vanilla | | 1.94 |
| | Qwen2-Audio-base | [12] | 3.08 |
| | Baichuan-Audio-base | [37] | 3.87 |
| AISHELL-2 ios [20] | Step-Audio-chat | [49] | 3.89 |
| | Qwen2.5-Omni | [59] | **2.56** |
| | VITA-Audio-Plus-Vanilla | | 3.29 |
| | Qwen2-Audio-base | [12] | 8.40 \| 7.64 |
| | Baichuan-Audio-base | [37] | 13.28 \| 10.13 |
| | Freeze-Omni | [56] | 13.46 \| 11.80 |
| | VITA-1.5 (S3) | [25] | 10.0 \| 8.40 |
| WenetSpeech [67] | LUCY (S1) | [28] | 10.42 \| 8.78 |
| *test-meeting \| test-net* | Step-Audio-chat | [49] | 10.83 \| 9.47 |
| | Qwen2.5-Omni | [59] | 7.71 \| **6.04** |
| | VITA-Audio-Turbo | | 23.97 \| 18.66 |
| | VITA-Audio-Vanilla | | 17.34 \| 13.45 |
| | VITA-Audio-Plus-Boost | | 9.38 \| 8.97 |
| | VITA-Audio-Plus-Vanilla (S1) | | **6.68** \| 6.59 |
| | VITA-Audio-Plus-Vanilla | | 7.12 \| 6.90 |

Table E5: More results on Spoken Question Answering (SQA) benchmarks. "S$x$" denotes the $x$-th training stage of speech models.

| Model | | #Params | Llama Question | | TriviaQA | | Web Question | | Mean | |
|---|---|---|---|---|---|---|---|---|---|---|
| | | | S → T | S → S | S → T | S → S | S → T | S → S | S → T | S → S |
| Proprietary Models | | | | | | | | | | |
| MinMo | [9] | 7B | 78.9 | 64.1 | 48.3 | 37.5 | 55.0 | 39.9 | 60.7 | 47.2 |
| Open-source Models | | | | | | | | | | |
| Moshi | [19] | 7B | 62.3 | 21.0 | 22.8 | 7.3 | 26.6 | 9.2 | 37.2 | 12.5 |
| GLM-4-Voice | [66] | 9B | 64.7 | 50.7 | 39.1 | 26.5 | **55.0** | 39.9 | 45.3 | 31.0 |
| LUCY (S2) | [28] | 7B | 59.6 | 51.0 | 23.2 | 18.2 | 26.6 | 18.2 | 36.5 | 29.1 |
| MiniCPM-o2.6 | [61] | 7B | - | 61.0 | - | 40.0 | - | 40.2 | - | 47.0 |
| Llama-Omni | [23] | 7B | - | 45.3 | - | 22.9 | - | 10.7 | - | 26.3 |
| VITA-Audio-Boost | | 7B | 68.7 | 60.3 | 30.5 | 29.3 | 32.9 | 30.4 | 44.0 | 40.0 |
| VITA-Audio-Vanilla | | 7B | 71.3 | 66.3 | 31.9 | 30.1 | 33.5 | 31.4 | 45.6 | 42.6 |
| VITA-Audio-Plus-Boost | | 7B | **76.3** | 64.6 | 43.6 | 39.5 | 44.2 | 40.0 | 54.7 | 48.0 |
| VITA-Audio-Plus-Vanilla | | 7B | 75.6 | **68.0** | **45.9** | **42.7** | 45.0 | **41.7** | **55.5** | **50.8** |

Table E6: We have tested VITA-Audio after aligning with ASR and TTS tasks on several text modality benchmarks. After the ASR and TTS alignment, the model's original text understanding capabilities were indeed affected. However, this might be due to the lack of large-scale, high-quality text data used during training. Interestingly, we observed a slight improvement in performance on GSM8K, which may be due to the fact that our training dataset included a significant amount of math-related data.

| Model | MMLU[31] | GSM8K[13] |
|---|---|---|
| Qwen-7B-Instruct | 74.22 | 80.06 |
| VITA-Audio-Plus-Vanilla | 66.92 | 80.14 |

Table E7: Generation time (ms) of the first audio segment under different inference modes in streaming inference. To enable more real-time speech generation, we progressively increase the number of steps in the flow matching model during streaming inference. The table shows the decoding time when the sampling step of the flow matching model is set to 1.

| Inference Mode | Audio Encoder | First Audio Token Chunk | Audio Decoder | Sum |
|---|---|---|---|---|
| VITA-Audio-Boost | 39 | 53 | 151 | 243 |
| VITA-Audio-Vanilla | 39 | 236 | 151 | 426 |

