# OpenReview forum: "VITA-Audio: Fast Interleaved Audio-Text Token Generation for Efficient Large Speech-Language Model"
_NeurIPS.cc/2025/Conference — NeurIPS 2025 poster_

### Official Review · Reviewer_opy1 · 2025-06-03

**Clarity:** 3
**Significance:** 3
**Originality:** 3
**Rating:** 5
**Confidence:** 4

**Summary:**

As speech LLMs and voice assistant technologies continue to advance, the number of forward passes required before producing the first speech output has become an increasingly important factor. The authors introduce a novel method called Multiple Cross-modal Token Prediction (MCTP) into an interleaving-based voice interaction model, which significantly reduces latency. At the 7B scale, the method achieves a 3 to 5 times faster generation speed compared to standard autoregressive decoding.

**Questions:**

Questions are in Weaknesses.

**Ethical Concerns:**

["NO or VERY MINOR ethics concerns only"]

**Final Justification:**

Since my score was already positive, I've decided to keep it unchanged.

**Limitations:**

yes

**Paper Formatting Concerns:**

I didn't find any issues in this paper.

**Quality:**

3

**Strengths And Weaknesses:**

# Strengths
1. Although MCTP bears some resemblance to text-based approaches like speculative decoding and MTP (as the authors acknowledge), its adaptation to audio token prediction based on their own alignment observations during speech generation is innovative and effectively reduces latency.

2. The utility of the proposed method and model is validated across a range of tasks.

3. The model and code have been open-sourced, making a meaningful contribution to the broader research community.

# Weaknesses and Questions

While I do not see any major shortcomings in this work, several technical questions came to mind:

1. In Table 3, it might have been helpful to include comparisons with a wider variety of voice interaction models. Currently, only parallel or interleaving-based speech LLMs are listed. Recent models like Qwen2.5-Omni, LLaMA-Omni, and MiniCPM-o incorporate pluggable decoders jointly trained with LLMs. Including results from these models would be informative. While they may show stronger performance due to better LLM preservation, I do not believe this would detract from the advantages of the proposed method, as the two approaches follow fundamentally different paradigms. I mention this only as a point of curiosity and not as criticism.

2. Since the authors mention using the decoder from glm-4-voice, I assume that the token format from that model is used at the output stage. In this context, I have two specific questions:
- (1) Does the Sensivoice encoder in the Plus model provide continuous embeddings or discretized tokens as input?
- (2) What kind of information do glm-4-voice tokens encode? Do they contain only semantic information, or do they also represent acoustic features? Given that one of the key advantages of an end-to-end pipeline is its ability to capture paralinguistic cues, the latter would be ideal.

3. I am curious whether the model was trained with multi-turn conversational data. That is, does it support multi-round interactions? From the Appendix, it appears to be single-round only, but I may have overlooked something.

4. I would also like to understand how many audio tokens are typically accumulated before audio is reconstructed. Since this number directly affects user-perceived latency, it would be useful to know the design trade-offs. If the chunk size is too small, audio quality may degrade due to glitches; if too large, the latency benefits of MCTP could be reduced. I realize this might be more closely related to the glm-4-voice decoder, and I raise this purely out of interest rather than as a critique.

5. Is there any evidence showing how much the original text understanding capabilities of the LLM are affected by training with interleaving or parallel generation schemes? It would be useful to see performance on benchmarks like MMLU to assess any potential trade-offs.

6. Since Qwen2.5-7B was used as the backbone, I am curious whether there was a particular reason for not using an Instruct-tuned version.

---

> ### Author Rebuttal · Authors · 2025-07-31
>
> We sincerely thanks for your positive recognition of our work. Our point-by-point responses to all your comments are provided below.
>
> ------
>
> **Q1: In Table 3, it might have been helpful to include comparisons with a wider variety of voice interaction models.**
>
> Thank you for your suggestion. We have consulted the results of models such as LLaMA-Omni [1] and MiniCPM-o [2] on the SQA task, and we will incorporate their results into the revised version of the paper.
>
> ------
>
> **Q2.1: Does the Sensivoice encoder in the Plus model provide continuous embeddings or discretized tokens as input?**:
>
> In the VITA-Audio-plus, the SenseVoice [3] encoder provides continuous embeddings as input.
>
> **Q2.2: What kind of information do glm-4-voice tokens encode?**:
>
> The GLM4-Voice [4] tokenizer is capable of encoding acoustic features such as intonation, stress, and rhythm.
>
> ------
>
> **Q3:  Does it support multi-round interactions?**
>
> VITA-Audio supports multi-turn conversations. In Stage 4, we trained using the VoiceAssistant-400K [5], which includes multi-turn audio dialogues. We will update the data format in the appendix in the revised version to avoid any misunderstandings.
>
> ------
>
> **Q4: How many audio tokens are typically accumulated before audio is reconstructed?**
>
> We found that GLM4-Voice struggles to encode very short audio tokens (e.g., 2 audio tokens). Based on our experience, typically 8 audio tokens are a suitable choice. This is also the reason we choose the interleaving ratio for the Boost and Balance inference modes. The Balance mode requires two forward passes to generate enough audio tokens for decoding, while the Boost mode only requires one forward pass.
>
> ------
>
> **Q5: Is there any evidence showing how much the original text understanding capabilities of the LLM are affected by training with interleaving or parallel generation schemes?**
>
> We have tested VITA-Audio after aligning with ASR and TTS tasks on several text modality benchmarks. After the ASR and TTS alignment, the model’s original text understanding capabilities were indeed affected. However, this might be due to the lack of large-scale, high-quality text data used during training. Interestingly, we observed a slight improvement in performance on GSM8K, which may be due to the fact that our training dataset included a significant amount of math-related data. We will include this result in the revised version.
>
>
> | Model | MMLU | gsm8k|
> |----------|----------|----------|
> | Qwen-7B-Instruct | 74.22 |80.06|
> | VITA-Audio-plus |   66.92 | 80.14|
>
>
> ------
>
> **Q6: Was there a particular reason for not using an Instruct-tuned version?**
>
> We used Qwen-7B-Instruct as the pre-trained text LLM. We will clarify this point in the revised version.
>
> ------
> [1] Fang Q, Guo S, Zhou Y, et al. Llama-omni: Seamless speech interaction with large language models.
>
> [2] Yao Y, Yu T, Zhang A, et al. Minicpm-v: A GPT-4v level mllm on your phone.
>
> [3] An K, Chen Q, Deng C, et al. Funaudiollm: Voice understanding and generation foundation models for natural interaction between humans and LLMs.
>
> [4] Zeng A, Du Z, Liu M, et al. Glm-4-voice: Towards intelligent and human-like end-to-end spoken chatbot.
>
> [5] Xie Z, Wu C. Mini-omni: Language models can hear, talk while thinking in streaming.

---

### Official Review · Reviewer_GWRd · 2025-06-27

**Clarity:** 2
**Significance:** 2
**Originality:** 2
**Rating:** 3
**Confidence:** 4

**Summary:**

This work proposes VITA-Audio, a spoken LM that can generate text and speech token simultaneously. They introduce a lightweight multiple cross-model token prediction modules to enable fast audio token generation. With the proposed module, VITA-Audio can generate 10 audio tokens after each LLM forward pass, mitigating the latency issue in other recent speech-text joint SLMs.

**Questions:**

1. How does the number 10 come from for the MCTP module to generate the audio tokens in each forward?

2. Figure 2 is unclear for me. Can you explain more about differences of the input of the MCTP modules and the spoken LM? Why is there only 5 audio token being fed into the decoder after each LLM’s forward? Do the MCTP modules only look at temporal hidden states instead of all historical output from the LLM? (related to weakness point 2, feel free to answer these questions and comments together)

3. What is the reason of having so many variants of the VITA-Audio model? The existence of the models increase the complexity of understanding or evaluating the actual performance of the model.

**Ethical Concerns:**

["NO or VERY MINOR ethics concerns only"]

**Final Justification:**

Overall, I would like to maintain my score due to:

1. The clarity issues would require a major revision.
2. The novelty is considered limited for me—integrating MCTP into the joint SLM framework appears to be a straightforward latency enhancement at the expense of performance (Boost/Turbo vs. Vanilla), and I found no particularly intriguing or surprising empirical results.

However, I do appreciate the fact that the authors decide to open-source the model, which should be one of the strengths that I have not mentioned in the review previously.

**Limitations:**

yes

**Quality:**

3

**Strengths And Weaknesses:**

**Strengths**

* The models exhibit reasonable latency-performance trade-off according to the result tables.

**Weaknesses**

* The writing style feels like a technical report rather than an academic paper. The problem is not well-formulated and the inputs / outputs of each modules are unclear.

* The process of how the speech-text data is interleaved is unclear in the main context. This affects the understanding of how the SLM is trained along with the MCTP modules. I have to check the supplementary materials to find out some clues and details (Figure E1 helps a lot). My recommendation is to make a clearer illustration that combines the advantages from both the Figure 2 and Figure E1, or at least include Figure E1 and present it in a more precise way. Anyway, I would like to know how exactly the text and audio tokens are interleaved in the training and inference stages. It seems that some hyperparameters are set to ensure that the audio tokens are lagged behind the text outputs during inference.

* The paper seems to be a incremental work that tries to mitigate the latency problem in GLM-4-Voice by introducing the concept in speculative decoding techniques. In this way, the novelty is considered limited for me. More importantly, the writing style should be revised towards being incremental rather than proposing an entire new type of SLM.

---

> ### Author Rebuttal · Authors · 2025-07-31
>
> Sincerely thanks for your efforts in reviewing this work. We hope the detailed responses help clarify your concerns. We would greatly appreciate it if you could kindly re-evaluate our work in light of the new explanations and additional results.
>
> ------
>
> **Q1: Choosing the Number of MCTP Modules:**
>
> The number of MCTP modules is positively correlated with inference acceleration; however, it may also come with an increased risk of error rate. After a comprehensive evaluation, we selected the configuration with 10 modules to achieve the optimal balance between efficiency and accuracy.
>
> ------
>
> **Q2: VITA-Audio Data Format.**
>
>
> We will take the inference stage as an example, and denote $Y_t$ as the $t$-th token, $h_n$ and $h^i_n$ as the $t$-th hidden state from LLM and MCTP-$i$, respectively.
>
> Without considering KV cache, LLM takes all previous $t$ tokens $( Y_0, Y_1, ..., Y_{t-1} )$ as input and predicts the next token $Y_{t}$. Note that LLM also generates hidden state $( h_0, h_1, ..., h_{t-1} )$.
>
> Then the first MCTP-1 futher takes $n$ tokens $( Y_1, Y_2, ..., Y_{t} )$ with the hidden states $( h_0, h_1, ..., h_{t-1} )$ as input and predict token $Y_{t+1}$. And MCTP-1 also generates hidden state $ h^1_{t} $ for token $Y_{t}$.
>
> The second MCTP-2 takes the all $n$ tokens $( Y_2, Y_3, ..., Y_{t+1} )$ with the hidden states $( h_1, h_2, ..., h_{t-1}, h^1_{t} )$ as input and predict token $Y_{t+2}$.
>
> After one forward pass of LLM and all MCTPs, the model outputs 11 tokens $( Y_{t}, Y_{t+1}, ..., Y_{t+10} )$.
> Figure 2 may cause some misunderstandings, and we will correct this in the revised version.
>
> During training, the loss function is formatted as:
>
> $ \mathcal{L}  = - \sum_{t=1}^{T} \log P \( Y_t  | {Y_0, Y_1, ..., Y_{t-1} } \)  + \log P_1 \( Y_{t+1}  | {Y_1, Y_2, ..., Y_{t},  h_0, h_1, ..., h_{t-1} } \)  + \log P_2 \( Y_{t+2}  | {Y_2, Y_3, ..., Y_{t+1},  h_1, h_2, ..., h^1_{t} } \)  + ... + \log P_{10} \( Y_{t+10}  | {Y_10, Y_11, ..., Y_{t+9},  h_9, h_{10}, ..., h^9_{t+9} } \)  $.
>
>
> We set the ratio of text tokens to be roughly $1:2$ following recent TTS and SLM literature (GLM4Voice [1] and CosyVoice2 [2]), which causes the audio stream to lag behind the text stream.
> During both inference and training, we use $n$ text tokens and $2n+2$ audio tokens as an interleaved group where the additional two tokens correspond to special audio tokens: <|begin_of_audio|> and <|end_of_audio|>.
>
> ------
>
>
> **Q3: The variants of VITA-Audio.**
>
> The variants of VITA-Audio come from two aspects:
>
> 1. Two Models with Different Architectures: VITA-Audio and VITA-Audio-Plus.
>
> VITA-Audio is designed as a universal paradigm to benefit various interleaved Speech LLMs.
> To validate the broad applicability of this framework, we conducted experiments on different architectures of interleaved speech LLMs.
> The results show that both VITA-Audio-Plus, based on SenseVoice, and VITA-Audio, based on GLM4-Voice, significantly improve the model's accuracy and inference speed by introducing the MCTP module, demonstrating the compatibility of this approach with different model architectures.
>
> 2. Four Different Inference Modes: Turbo, Boost, Balance, and Vanilla.
>
> We have designed four different inference modes: Turbo, Boost, Balance, and Vanilla, to be applied in different scenarios.
> The core distinction between these modes lies in the configuration of inference parameters (the number of activated MCTP modules and the ratio of interleaved text and audio token generation).
> As a result, we do not need to train a separate model for each mode.
> Of course, targeted optimization can still be carried out for the desired inference mode if the application scenario is specified.
>
>
> ------
>
>
> **Q4: Differences Between MCTP and Speculative Decoding:**
>
> Existing speculative decoding methods (such as Medusa[3]) typically require verifying all candidate tokens to ensure the correctness of the generated results.
> And the verification step significantly slows down the decode speed.
> If the verification step is omitted, the accuracy rate of the generated results tends to be low.
> In high-concurrency or resource-limited environments, the acceleration benefits of speculative decoding diminish significantly.
>
> In contrast, the MCTP module aims to reduce the latency for generating the first audio chunk in streaming scenarios, and does not require verification.
> Our instantiated MCTP generates up to ten audio tokens at once.
> More importantly, this module has a very low computational cost, ensuring stable and significant interaction acceleration even in high-concurrency or resource-constrained settings.
>
> It is important to emphasize that speculative decoding methods and MCTP are not mutually exclusive.
> In practical applications, speculative decoding can be used to accelerate text generation, while the MCTP module can be employed during speech generation, further improving overall end-to-end inference speed.
>
> ------
>
> [1] Zeng A, Du Z, Liu M, et al. Glm-4-voice: Towards intelligent and human-like end-to-end spoken chatbot.
>
> [2] Du Z, Wang Y, Chen Q, et al. Cosyvoice 2: Scalable streaming speech synthesis with large language models
>
> [3] Cai T, Li Y, Geng Z, et al. Medusa: Simple LLM inference acceleration framework with multiple decoding heads.

---

> ### Comment · Reviewer_GWRd · 2025-08-04
> **Final Justification**
>
> Thank you for the clarifications. Overall, I would like to maintain my score due to:
> 1. The clarity issues would require a major revision.
> 2. The novelty is considered limited for me—integrating MCTP into the joint SLM framework appears to be a straightforward latency enhancement at the expense of performance (Boost/Turbo vs. Vanilla), and I found no particularly intriguing or surprising empirical results.
>
> However, I do appreciate the fact that the authors decide to open-source the model, which should be one of the strengths that I have not mentioned in the review previously.

---

### Official Review · Reviewer_orX3 · 2025-07-02

**Clarity:** 4
**Significance:** 4
**Originality:** 4
**Rating:** 4
**Confidence:** 5

**Summary:**

Problem & Motivation:

Traditional cascaded speech systems (ASR → LLM → TTS) suffer from cumulative latency, loss of paralinguistic cues, and error propagation. Even recent end-to-end speech-LLM models incur high first-token delays in streaming scenarios, which hinders real-time interaction. VITA-Audio addresses this by enabling the model to output audio tokens in the very first forward pass, eliminating that bottleneck.

**Questions:**

Can the authors provide a translation for all non-English characters to ensure everyone can understand the examples in Figure 1?

**Ethical Concerns:**

["NO or VERY MINOR ethics concerns only"]

**Limitations:**

Complex, Multi-Stage Training Pipeline: The four-stage curriculum (audio–text alignment → single-module training → multi-module training → fine-tuning) requires careful scheduling, hyperparameter tuning, and large amounts of pretraining data. This adds significant engineering overhead and may be hard to adapt to new domains.
However, this should not be a concern for the overall ratings.

**Quality:**

4

**Strengths And Weaknesses:**

Strengths:

- Zero Audio Token Delay: VITA-Audio is the first end-to-end speech model that can generate audio tokens in the very first forward pass.

- Interleaved Cross-Modal Token Prediction (MCTP): The lightweight MCTP modules efficiently map LLM hidden states to multiple audio tokens in one pass.

Weakness:

A central observation and motivation in VITA-Audio is exactly that “the hidden states of certain text tokens in the LLM backbone contain sufficient semantic information for generating the corresponding audio tokens, which means that it is unnecessary to attend to additional text tokens when generating audio.” I really like this interesting concept and the findings based on this, and also appreciate the astonishing results they achieved based on this characteristic. However, the paper doesn’t offer a full study or a preliminary experiment on an evaluation set with quantitative results to support this observation. Without this, it is not clear to the audience whether this is just a coincidence or is really a common phenomenon. Of course, I am definitely aware that the authors did mention “This finding is also reported in many literature on attention-based speech systems [7, 32]”, but the whole paper can still be improved by simply providing quantitative results based on the speech model they are using to support the hypothesis “irrelevant text tokens being masked out, the model is still able to generate the correct audio, and the pronunciation remains contextually appropriate”. Any statistical analyses or metrics would substantially strengthen the claim.
I will uprate my score if this concern is addressed.

---

> ### Author Rebuttal · Authors · 2025-07-31
>
> We sincerely thanks for your positive recognition of our work. Our point-by-point responses to all your comments are provided below.
>
>
> ------
>
> **Q1: Can the authors provide a translation for all non-English characters to ensure everyone can understand the examples in Figure 1?**
>
> Thank you for your suggestion.
>
> In Chinese, "一行" (yihang) emphasizes spatial arrangement (e.g., 一行白鹭 - a row/line of egrets), while "一行" (yīxíng) focuses on the concept of a group or unit (e.g., 一行人 - a group/party of people).
>
> In the revised version, we will provide translation annotations for all non-English characters to ensure smooth understanding for the readers.
>
> ------
>
> **Q2: Detailed supporting results for the hypothesis that, with irrelevant text tokens being masked out, the model is still able to generate the correct audio, and the pronunciation remains contextually appropriate.**
>
> Although we have not quantitatively tested this observation on a specific benchmark, we have verified it through several cases. During inference, we first locate the audio tokens that the model attends to most in the attention map for a given text token, and then perform text-token masking before generating those audio tokens. Specifically, we generate audio tokens with three different masking strategies:
>
> (1) No text token is masked at all.
>
> (2) All irrelevant text tokens except the target text tokens are masked.
>
> (3) All text tokens are being masked.
>
> We then continue the generation after the masking. In this case, both (1) and (2) can produce normal speech.
>
> For examples in Figure 1a), in the phrase "there are seven days in a week," the attended text is on "seven days," and in the phrase "一行白鹭上青天," the attended text is on "一行" (yi hang), meaning "One row of egrets ascends the blue sky," and both can produce correct pronunciation.
>
> We are conducting quantitative experiments. Specifically, we plan to perform forced alignment on the generated speech to obtain the corresponding audio tokens corresponding to each text token. We then regenerate the speech and mask out all text tokens except for the relevant ones when reaching those audio token positions. We will assess whether the generated audio tokens are correct. Unfortunately, due to time constraints, we don't have the final result yet, but we will synchronize the results once they are available.

---

> > ### Comment · Reviewer_orX3 · 2025-08-05
> >
> > I have read the rebuttals submitted by the authors. I decided to maintain my score since there are no quantitative results to support the statement "the hidden states of certain text tokens in the LLM backbone contain sufficient semantic information for generating the corresponding audio tokens, which means that it is unnecessary to attend to additional text tokens when generating audio"

---

### Official Review · Reviewer_4oP8 · 2025-07-03

**Clarity:** 3
**Significance:** 2
**Originality:** 2
**Rating:** 4
**Confidence:** 4

**Summary:**

This paper introduces VITA-Audio, an end-to-end speech model designed for real-time audio generation during the first forward pass, addressing latency issues in cascaded and interleaved generation pipelines. The core innovation lies in the MCTP module, which allows the model to generate decodable audio token chunks with zero audio token delay, achieving 3-5x inference speedup while maintaining high-quality output. VITA-Audio is trained solely on open-source data, fully open-sourced, and evaluated comprehensively on ASR, TTS, and speech question answering benchmarks. Results show that VITA-Audio is competitive with existing models of similar size in both efficiency and accuracy, while offering low-latency generation.

**Questions:**

- MCTP novelty: The MCTP module is central to your approach. Could you clarify how it differs from existing chunked or multi-head generation strategies commonly used in LLMs and TTS systems? For instance, could LoRA adapters or other parameter-efficient methods be used instead of MCTP modules to achieve similar latency and throughput benefits?

- Multispeaker synthesis: How is multispeaker TTS handled in VITA-Audio? Are speaker embeddings used in the decoder, and if so, how are they obtained and integrated during training?

- Training stages: The current setup involves multiple stages. Are all of these steps necessary for good performance? For example, could the MCTP modules be trained directly in Stage 1 while fine-tuning the LLM, without needing the full staged pipeline?

**Ethical Concerns:**

["NO or VERY MINOR ethics concerns only"]

**Final Justification:**

The rebuttal clarifications addressed my concerns. While the novelty is limited, I believe it is fair and the contribution is solid. The paper is clearly written, reproducible, and evaluated thoroughly. Therefore I confirm my initial score.

**Limitations:**

Yes

**Quality:**

2

**Strengths And Weaknesses:**

**Strengths**

- The paper proposes a simple and practical solution to reduce token latency in interleaved speech-language models, which is important for enabling more human-like, real-time conversational systems.

- The model is fully open-source and reproducible, trained entirely on public datasets.

- The evaluation is comprehensive, covering multiple tasks (ASR, TTS, SQA) and speech language models.

- The paper is clearly written and well-structured.


**Weaknesses**

- Methodological novelty is limited. The work builds on known ideas such as the fact that speech-text alignment is many-to-one, and adds incremental improvements (e.g., MCTP, adapter-based tuning). While useful, these do not represent a major modeling innovation.

- The evaluation focuses mainly on semantic modeling. There is no acoustic modeling, which limits the system's ability to handle speaker consistency, prosody, or low-level audio fidelity. Tasks like speaker-consistent speech continuation are likely unsupported. Comparisons with acoustic-semantic models (e.g., Moshi) may be unfair, as VITA-Audio only models semantic tokens.

- The paper does not compare against other methods to for parallel tokens generation (e.g., LoRA adapters, Medusa, speculative decoding), which could offer similar benefits in terms of latency compared to using MCTP modules.

**Overall Assessment**

VITA-Audio is a good engineering effort that advances real-time speech generation using open-source models and data. Although the methodological novelty is modest, the model is efficient, well-evaluated, and reproducible. It offers a practical approach to low-latency, semantic token-based speech generation.

---

> ### Author Rebuttal · Authors · 2025-07-31
>
> We sincerely thanks for your positive recognition of our work. Our point-by-point responses to all your comments are provided below.
>
> ------
>
> **Q1: Innovation of the MCTP Module.**
>
> 1. Design Motivation: The design motivation for the MCTP module stems from our observation that the hidden states of certain text tokens in the LLM backbone contain sufficient semantic information for generating the corresponding audio tokens. Based on this, we aim to design a lightweight module capable of efficiently generating audio tokens from these hidden states. Potential candidates for this lightweight module include: a set of Transformer layers, a small-scale LLM, or reusing the final layers of the LLM by feeding the last-layer hidden states back to those layers. Ultimately, for the simplicity of the training pipeline, we choose to use a set of sequential Transformer layers as the core of the MCTP module.
>
> 2. Advantages over Speculative Decoding: Existing speculative decoding methods (such as Medusa[1]) typically require verifying all candidate tokens to ensure the correctness of the generated results. If the verification step is omitted, the accuracy rate of the generated results tends to be low. Even with optimizations like tree attention, the verification phase still requires inputting longer candidate sequences into the LLM for forward computation, and the input sequence length increases rapidly as the number of parallel predicted tokens grows. In high-concurrency or resource-limited environments, the acceleration benefits of speculative decoding diminish significantly. In contrast, the MCTP module does not require verification and can generate up to ten audio tokens at once. More importantly, this module has a very low computational cost, ensuring stable and significant inference acceleration even in high-concurrency or resource-limited settings.
>
> 3. Complementarity: It is important to emphasize that speculative decoding methods and MCTP are not mutually exclusive. In practical applications, speculative decoding could be used to accelerate text generation, while the MCTP module can be employed during speech generation, further enhancing the overall end-to-end inference speed.
>
> ------
>
> **Q2: Regarding Multispeaker Synthesis.**
>
> Currently, our main focus remains on ASR, TTS, and SQA tasks. Although speaker embeddings are used in the audio decoder, we have not yet conducted specialized training for such tasks.
>
> ------
>
> **Q3: Training method.**
>
> Our training method is divided into four stages, primarily based on the following two considerations:
>
> 1. Our goal is to make VITA-Audio a universal paradigm that can benefit all interleaved Speech LLMs. Specifically, the model can directly leverage the weights aligned during ASR and TTS tasks to quickly initiate the training of the second stage. To validate the generalization ability of the framework, we trained two versions, VITA-Audio and VITA-Audio-Plus, to demonstrate the effectiveness of this approach across different model architectures.
>
> 2. In early experiments, we attempted to train the MCTP module directly in Stage 1, or train a version containing 10 MCTPs in Stage 2, all at once. However, when the training data was insufficient, such approaches performed worse than the staged training strategy.
>
> ------
>
> [1] Cai T, Li Y, Geng Z, et al. Medusa: Simple LLM inference acceleration framework with multiple decoding heads.

---

> ### Comment · Reviewer_4oP8 · 2025-08-04
>
> Thank you for your rebuttal. Your clarifications help address most of my concerns. I now feel confident that my initial positive assessment was fair and will maintain my original score.

---

### Note · Authors · 2025-08-13

We appreciate the constructive and encouraging feedback from the four reviewers.

**Reviewer 4oP8** expressed concerns regarding the novelty of the MCTP module and necessity of the 4-stage training. After we provided further clarification, Reviewer 4oP8 stated that **"clarifications help address most of my concerns"** and also noted that **"feel confident that my initial positive assessment was fair."**

**Reviewer orX3** requested quantitative results and mentioned that **addressing this concern would lead to a higher score**. Although we were unable to conduct the experiment earlier due to time and resource constraints, **we have now supplemented our work with a quantitative result that supports our hypothesis.**

We first collect 50 sentences containing polyphonic characters and generate the speech by masking some text tokens and evaluate the accuary of the generated speech for the polyphonic characters.
|Masking Strategy|Correct samples|
|-|-|
|No masking|49|
|Mask irrelevant text tokens|46|
|Mask all text tokens| 0|

As shown, in the most cases, even when all irrelevant text tokens are masked except for the text tokens of polyphonic characters, the model is still able to produce correct pronunciations. This result strongly supports the our hypothesis. **We will also release the testing scripts to the community for reproduction.**

**Reviewer GWRd** commented that MCTP is a straightforward latency enhancement at the expense of performance.
VITA-Audio enables zero-delay generation of decodable audio token chunks and achieves nearly 5× inference speedup at the expense of only a slight performance drop—from 50.8% to 48.0% as shown in Table 3—which still outperforms other models. It is important to emphasize that speculative decoding and MCTP are compatible and can be combined to further boost overall end-to-end inference speed.

In addition, Reviewer GWRd **greatly appreciated our decision to open-source the model, recognizing it as a valuable contribution to the development of the community.**

**Reviewer opy1** considered our work to be "making a meaningful contribution to the broader research community." After discussing several technical questions with us, the reviewer indicated that they **"will maintain my original score."**

We sincerely thank all reviewers for their constructive feedback and valuable suggestions. We will incorporate these recommendations in our revision, with the aim of making further contributions to the community.

---

### Decision · Program_Chairs · 2025-09-17

**Decision:**

Accept (poster)

**Comment:**

VITA-Audio is an end-to-end speech model designed for real-time audio generation during the first forward pass, addressing latency issues in cascaded and interleaved speech systems.

Strengths:

•	Low-latency generation: VITA-Audio achieves 3–5× speedup over standard autoregressive models, making it suitable for real-time applications.

•	MCTP module: Efficiently maps LLM hidden states to multiple audio tokens in one pass, reducing delay.

•	End-to-end design: Avoids cascading ASR → LLM → TTS pipelines, mitigating error propagation and preserving paralinguistic cues.

•	Open-source commitment: Model, training code, and data are publicly released.

•	Comprehensive evaluation: Benchmarked on ASR, TTS, audio language modeling, and spoken question answering tasks.

Overall Reviewers’ attitudes:

Most of reviewers generally appreciate the practical impact, efficiency, and open-source nature of VITA-Audio. While some see the contribution as solid but incremental, others suggest that clarity improvements and stronger empirical validation would significantly enhance the paper. The model’s ability to generate speech in the first forward pass is seen as a notable advancement for real-time voice interaction systems.